# Effect of β-Mannanase Addition during Whole Pigs Fattening on Production Yields and Intestinal Health

**DOI:** 10.3390/ani12213012

**Published:** 2022-11-02

**Authors:** Pedro Sánchez-Uribe, Eva Romera-Recio, Carolina G. Cabrera-Gómez, Elisa V. Hernández-Rodríguez, Álvaro Lamrani, Belén González-Guijarro, Clara de Pascual-Monreal, Livia Mendonça-Pascoal, Laura Martínez-Alarcón, Guillermo Ramis

**Affiliations:** 1ELANCO Animal Health, 20108 Alcobendas, Spain; 2Estación Experimental del Zaidín (CSIC), 18008 Granada, Spain; 3Departamento de Producción Animal, Facultad de Veterinaria, Universidad de Murcia, 30100 Murcia, Spain; 4Escola de Veterinária e Zootecnia, Universidade Federal de Goiás, Goiánia 74690-900, Brazil; 5Instituto Murciano de Investigación en Biomedicina (IMIB), 30100 Murcia, Spain; 6UDICA, Hospital Clínico Universitario Virgen de la Arrixaca, 30100 Murcia, Spain

**Keywords:** pathogen associated molecular patterns, β-mannans, β-mannanase, finishing, growing, intestinal integrity, pigs

## Abstract

**Simple Summary:**

Raw materials used in the manufacture of pig feed may contain anti-nutritional elements. These include β-mannans: oligosaccharides that produce a state of unnecessary inflammation in the intestine, hindering the absorption of nutrients and worsening animal performance. In this trial, an enzyme that degrades these sugars was used throughout the growing–finishing period, also reducing the energy of this feed (HC), compared to a control feed (CON). The animals were weighed, and growth and FCR were calculated. In addition, fecal consistency, gastric lesions at slaughter and a battery of 16 biomarkers in feces and tissues, indicators of intestinal integrity and immune stimulation, were studied. HC animals grew as well as CON animals and had a lower FCR. In addition, an anti-inflammatory state was observed in feces and in jejunum and ileum tissue at slaughter, suggesting that the use of this enzyme effectively controls the β-mannan-derived immune reaction.

**Abstract:**

The presence of β-mannans in feed can produce a futile and chronic immune stimulation in fattening pigs. In this trial, a 1-4-endo-D-β-mannanase was added to the feed (HC) during growth and fattening (0.03% of Hemicell HT) and physical performance and pathological data were recorded, and intestinal integrity and immune activation were studied by molecular biomarkers, compared to a control group (CON). The treatment diet was reduced in energy content by 40 Kcal/kg NE. From each group, 113 and 112 animals housed in 8 pens were individually identified and weighed three times: at 7th, 63rd and 116th days in feed. The FCR was calculated for groups of two pens and ADG individually. There was no difference in ADG (CON = 0.836, HC = 0.818) nor in FCR between groups (*p* = 0.486). During growth, there was a higher frequency of normal feces in HC and there were also no differences in the frequency of gastric lesions. A significant increase in Claudin, Occludin, IFN-γ and IL8 was observed in the CON in feces and a significant decrease in IL-6 in HC. In tissues, there were differences for IL-12p40, TNF-alpha in jejunum (increased CON) and TGF-β in ileum and jejunum, (decreased HC). The economic performance was EUR 4.7 better in the treated group. In conclusion, the addition of 1-4-endo-D-β--mannanase to the feed with a 1.6% reduction in net energy compared to the control, allowed the animals to perform as well as the animals on the higher energy diet, with lower prevalence of diarrhea.

## 1. Introduction

The ban of the use of zinc oxide in 2022 [1] and the reduction of antibiotic use in enteric disease prophylaxis in the European Union [2], opens a bleak outlook for piglet intestinal health, especially post-weaning. Zinc oxide has been shown to be highly effective in reducing the frequency of post-weaning diarrhea in piglets and improving productive performance [3], when used at therapeutic doses up to 3000 ppm, through stabilization of the microbiota, preventing the adhesion of pathogenic bacteria to enterocytes and reducing inflammation and oxidative stress [4]. However, its massive use can lead to environmental disturbances due to the accumulation of this metal where slurry with a high concentration of this metal is used as an organic amendment. Its use has also been linked to the emergence of antibiotic-resistant strains of *Escherichia coli* [5]. Therefore, it is currently only authorized for use in feed in amounts up to 150 ppm, in accordance with standard nutritional recommendations and current EU legislation [6].

Compounding this effect, in 2022 there is a raw material crisis due to the European geopolitical situation that has led to lower quality standards for imported raw materials for the manufacture of animal feed [7], which will probably result in a worsening of the quality of animal feed from the animal health standpoint. We must take into account that some raw materials contain elements called anti-nutritional factors that result in a loss of intestinal health. One of the main ones are non-starch polysaccharides (NSP), high molecular weight complex carbohydrates that reduce digestion efficiency and yields in all production species. Among them are the β-mannans (glucomannan, galactomannan, and galactoglucomannan), which are heat-resistant and found in dehulled soybeans [8]. These elements are not only found in soybean hulls but can also be found in different proportions in common raw materials such as barley, copra meal, DDGS, palm kernel meal, rice or sunflower meal [8,9]. In weaned piglets, substitution of soybean meal with concentrated soy protein concentrate, fish meal, or enzyme-treated soybean meal has been shown to improve growth, reduce the frequency of diarrhea and improve duodenal villus morphology [10].

These β-mannans induce an energy-costing futile immune response that prevents nursery pigs from reaching maximum growth potential. Vertebrates, such as pigs, have an evolutionarily preserved innate immune system that recognizes lipopolysaccharides and mannans of yeast as microbe-associated molecular pattern molecules, which have classically been referred to as having a pathogen associated molecular pattern [11]. One of the most used strategies is the substitution of raw materials such as soybean meal for others such as protein concentrate or subjecting soybeans to various treatments [12,13], but this makes the cost of the diet more expensive, so it is necessary to look for strategies which cost makes the feed cheaper or makes it possible to maintain the level of incorporation of materials such as soybean meal. To cope with the effect of β-mannans, different strategies have been developed, especially in chickens, including the use of enzymes such as endo-1,4-β-mannanase. This enzyme hydrolyzes β-mannans such that they reduce the immune challenge and thus allow for dietary energy to be used for growth rather than immune response [14].

On the other hand, there has been a lot of interest in investigating the effect of different factors that can cause intestinal dysfunction, chronic inflammation or what is known as weak gut after weaning, as this is a critical time in the life of the animals [15,16]. However, the entry into fattening is another moment of great stress for the piglets and in addition, given the duration of the fattening period, and the fact that the last part of the productive life of the animals is where the return on capital must occur, it seems interesting to investigate the factors that could reduce the yields in pig production with reduced antibiotic use. Factors that cause alterations in the intestinal integrity and nutrient absorption of post-weaned piglets, such as transport [17] or heat stress [18], are known to produce the same effect in fattening pigs, at the beginning of the fattening period or even at the end of the finishing. Hence, poor gut health and chronic low-level gut inflammation are important topics for optimal antibiotic-free poultry and pigs’ production [19] and should be carefully investigated.

It has been hypothesized that the addition of endo-1,4-β-mannanase to the feed throughout the fattening period would enhance the effect of β-mannan on the immune system, reducing futile inflammation in the intestine and improving production performance to the point of allowing reduce the energy content of the feed with the associated cost savings. To test the hypothesis, the following objectives were set: to know the growth and feed efficiency during fattening, as well as the activation status of genes related to intestinal integrity, the presence of inflammatory cells and the state of local immune stimulation. The frequency of diarrhea, the use of drug treatments and the state of stomach lesions at slaughter were also investigated as indicators of health. All these terms have been investigated under commercial and non-experimental conditions, which are the conditions of most of the studies published in the literature.

## 2. Materials and Methods

### 2.1. Farm and Animals

The trial was conducted in a piggery, placed in the province of Murcia (S-E Spain; 37°41′43″ N 1°58′54″ W), oriented NNE-SSW. Each piggery was loaded with 510 piglets during 24 to 27 September 2021, all coming from the same origin farm. The finishers were emptied on 3 February 2022. As an AI/AO protocol was applied, the load of piggery 1 was started until full (24 September 2021), followed by the load of piggery 2 (27 September 2021). The treatment was randomly assigned to each piggery, with the control feed being taken from piggery 1 (control group; CON) and the treated feed from piggery 2 (treated group; HC). During the entire fattening period, temperature measurements were taken every 10 min by continuous recording datalogger (Maxim Integrated, San José, CA, USA), as an indicator of environmental quality and no differences were obtained between the two piggeries in daily minimum temperature (18.2 ± 0.3 °C and 17.86 ± 0.3 °C for piggeries 1 and 2, respectively), daily maximum temperature (23.86 ± 0.3 °C and 23.58 ± 0.3 °C, respectively) or daily mean temperature (20.91 ± 0.3 °C and 20.52 ± 0.2 °C, respectively). The average difference between daily minimum and daily maximum temperature was 5.56 ± 0.24 °C and 5.72 ± 0.26 °C for piggery 1 and 2, respectively, (not significant difference) indicating good environmental control of temperature.

The 3 × 3.5 m pens housed 14–15 animals, with a two-places concrete feeder, shared by two adjacent pens, and a nipple-drinker with water ad libitum. Feed was offered ad libitum throughout the fattening period. Each experimental group was set up with half males and half females, with the same number of pens of each sex (4 males and 4 females in each piggery).

All the experimental procedures described in this research were carried out under the welfare rules stated in R.D. 1135/2002 for the protection of pigs, and at no time any procedure involving more pain than the insertion of a needle was performed. Anyway, the trial was approved by the Bioethical Committee of the University of Murcia (CEEA-OH 465/2018).

### 2.2. Feed

Two feeds were formulated, using the same raw materials and ensuring at least 12% soybean meal incorporation. The composition of the feeds is shown in Table 1. The analysis of macronutrient and energy values is given in Table 2. The feed formulation was made to reduce by 40 Kcal/kg NE in the treated diet.

The treated feed was supplemented with 0.03% Hemicell HT (Elanco AH, Indianapolis, IN, USA), containing endo-1,4-β-mannanase produced by fermentation of *Paenibacillus lentus.* The amount of Hemicell HT was sufficient to ensure a minimum content of 48,000 U of enzyme/kg of diet.

The cost of the treatment is included in the final cost of the feed.

The content of β-mannans was calculated as 0.42% for Barley, 0.14 for corn, and 0.59 for soya meal 48%. So, the β-mannan content of the diets used was rated as at least >0.225% in any of the feeds, using the tool provided by ELANCO AH.

### 2.3. Performances

To record individual weights (BW), growth (ADG) and Feed Conversion Rate (FCR), 113 animals were individually ear-tagged in each piggery; housed in 8 pens, which shared a feeder two by two. The CON group included 57 males and 56 females, and the HC group included 57 males and 55 females.

The animals were weighed three times; W1 at 10th (piggery 1) and 7th (piggery 2) day after finishing commenced; W2 at the end of growing (day 63 and 60 on feed) and W3 before the first batch to slaughter (116 and 113 days on feed). The number of animals weighed is shown in Table 3.

The weight increase was calculated during growing (WG1), during finishing (WG2) and during the whole fattening period (WGT). The average daily gain during growth (ADG1), during finishing (ADG2) and during the whole period (ADGT) was also calculated.

The animals which weight was interpolated were included in the analysis of feed conversion rate (FCR) and for total weight gain (WGT) but not for WG or ADG calculations.

The animals were individually weighed in an electronic scale ETW VA with a weight indicator BWI 30,000 (Bosche GmbH & Co., KG, Damme, Germany).

The diet supplied to each group of two pens was weighed on a two days basis to determine the feed intake of each group of blocks in each piggery, using a Platform Scale IPS-B (Bosche GmbH & Co., KG, Damme, Germany). The pens were placed at the two ends of each piggery to avoid phenomena resulting from the positioning of the animals. The Average Daily Feed Intake was calculated on a pen basis (ADFI) and on an individual basis (MADFI).

The first batch was slaughtered at 117th and 124th days in feed for piggery 1 and 2, respectively, and the finishing was end at 132nd and 138th days on feed. The average days in feed were 124 and 130 for CON and HC, respectively.

### 2.4. Clinical Observation

#### 2.4.1. Clinical Status and Treatments

Throughout the fattening period, the clinical status of the animals housed in the two piggeries was observed, with special attention to the animals housed in the 8 monitored pens in each piggery. Individual parenteral treatments were recorded on a daily basis, recording the identity of the animal administered, the pen in which it was housed, and the drug administered. The administration of systemic medications via feed or water to the whole group was also recorded, which was decided under the criteria of the clinical veterinarian responsible for the farm. In this case, the duration, the prescribed drug and the route of administration (via water or feed) were also recorded.

#### 2.4.2. Fecal Score

The fecal score in a pen basis were observed every two days during finishing, using a 0 to 3 score system: 0, normal feces, 1 pasty, 2 creamy and 3 liquids, according to a score scale previously described [20]. The score was assigned to the highest score observed regardless of the number of animals showing this score.

### 2.5. Biomarkers for Intestinal Integrity and Immune Stimulation

#### 2.5.1. Sampling

Fecal samples were obtained during the second and third weight. A sample of 100 mg was included in an Eppendorf containing 1 mL of RNAlater (Invitrog, Waltham, MA, USA) to preserve RNA; after 24 h in refrigeration were frozen at −80 °C up to analysis.

At slaughter, 20 samples of ileum, jejunum and colon were divided; a 100 mg portion was included in RNAlater and a bigger piece was fixed in formalin for histopathology. The animals were randomly selected among those in the first batch sent to abattoir.

#### 2.5.2. Gene Expression

Gene expression for cytokines IL-1α, IL-1β, IL-6, IL-8, IL-10, IL-12p35, IL-12p40, TNF-α, IFN-α, IFN-γ, and TGF-β was determined by means of relative quantification, using primers previously described by various authors (Table 4). Moreover, gene expression for cell infiltration and tight junction integrity was assessed by means of calprotectin (CAL), occludin (OCL), zonulin-1 (ZON) and claudin (CLAU) quantification. The primers are shown in Table 5, as previously described, except for CLAU which was designed by means of PrimerBlast (https://www.ncbi.nlm.nih.gov/tools/primer-blast/ (accessed on 20 September 2022)) using the sequency recorded in the Table. Since there were two fecal samplings, for correlation analysis the biomarkers were assigned as 1 (63rd days in feed) or 2 (116th day in feed). The biomarkers analyzed at slaughter were marked as S. So, for instance could be IFNγ1, IFNγ2 and IFNγS, depending on the samples analyzed.

For q-PCR develop, briefly, total RNA was isolated from 100 mg feces or 20 mg of tissue samples and PBMC by using the Thermo Scientific Gene JET RNA Purification Kit (ThermoFisher, Waltham, MA, USA) and cDNA was synthetized using the Geneamp RNA PCR Core Kit (Life Technologies, Carlsbad, CA, USA) using oligo-dT as primers to get cDNA only from mRNA. The PCRs were performed using a 7300 ABI thermocycler (Life Technologies, Carlsbad, CA, USA) and the GoTaq q-PCR Master Mix (Promega, Madison, WI, USA) with SYBR-Green chemistry. The specificity of the reaction was assessed by analyzing the melting curve. The samples were normalized using the Ct for β-actin. The expression for each sample was calculated [21], correcting to the PCRs efficiency, which was calculated by serial decimal dilutions and using the slope offered by the thermal cycler software, and used as the control group for the CON animals. Data were expressed as fold change, normalized to the lowest value (which was assigned a value 1), and the log2 was calculated to be able to compare the different biomarkers.

**Table 4 animals-12-03012-t004:** Primers for cytokines IL-1β, IL-6, IL-8, IL-10, IL-12p35, IL-12p40, TNF-α, IFN-α, IFN-γ, and TGF-β and primers for β-Actin.

Gene	Primer Forward (5′ → 3′)	Primer Reverse (5′ → 3′)	References
IFN-α	5′-CCCCTGTGCCTGGGAGAT-3′	5′-AGGTTTCTGGAGGAAGAGAAGGA-3′	[22]
IFN-γ	5-TGGTAGCTCTGGGAAACTGAATG-3′	5′-GGCTTTGCGCTGGATCTG-3′	[23]
TNF-α	5′-ACTCGGAACCTCATGGACAG-3′	5′-AGGGGTGAGTCAGTGTGACC-3′	[24]
IL-12p35	5′-AGTTCCAGGCCATGAATGCA-3′	5′-TGGCACAGTCTCACTGTTGA-3′	[22]
IL-12p40	5′-TTTCAGACCCGACGAACTCT-3′	5′-CATTGGGGTACCAGTCCAAC-3′	[25]
IL-10	5′-TGAGAACAGCTGCATCCACTTC-3	5′-TCTGGTCCTTCGTTTGAAAGAAA-3′	[23]
TGF-β	5′-CACGTGGAGCTATACCAGAA-3′	5′-TCCGGTGACATCAAAGGACA-3′	[22]
IL-8	5’-GCTCTCTGTGAGGCTGCAGTTC-3’	5′-AAGGTGTGGAATGCGTATTTATGC-3′	[26]
IL-1α	5’-GTGCTCAAAACGAAGACGAACC-3’	5′-CATATTGCCATGCTTTTCCCAGAA-3′	[27]
IL-1β	5’-AACGTGCAGTCTATGGAGT-3’	5′-GAACACCACTTCTCTCTTCA-3’	[28]
IL-6	5′-CTGGCAGAAAACAACCTGAACC-3’	5′-TGATTCTCATCAAGCAGGTCTCC-3’	[28]
β-actin	5’-CTACGTCGCCCTGGACTTC-3’	5’-GATGCCGCAGGATTCCAT-3’	[29]

**Table 5 animals-12-03012-t005:** Primers for tight junction (OCL, ZON and CLAU) and Calprotectin (CAL).

Gene	Forward Primer	Reverse Primer	References or Accession Number
CALPROTECTIN (S100 calcium binding protein A8)	5’-AATTACCACGCCATCTACGC-3′	5′-TGATGTCCAGCTCTTTGAACC-3′	[30]
*Occludin*	5′-TTGCTGTGAAAACTCGAAGC-3′	5′-CCACTCTCTCCGCATAGTCC-3′	[30]
*Zonulin 1*	5′-CACAGATGCCACAGATGACAG-3′	5′-AGTGATAGCGAACCATGTGC-3′	[30]
*Claudin 1*	5′-ACCCCAGTCAATGCCAGATA-3′	5′-GGCGAAGGTTTTGGATAGG-3′	MK452762.1

### 2.6. Gastric Lesions Score

At abattoir, 95 stomachs from each group were evaluated; observing the aglandular mucosa after opening the stomach by greater curve and using the score proposed by Ramis et al. [31], from 0 (normal) to 7 (Ulcer), as described in Table 6.

It was recorded whether the ulcers observed were acute, chronic, healed or crateriform.

A summary of the methods used in this trial is shown in Figure 1.

### 2.7. Statistical Analysis

Data collected in an Excel database (Microsoft Inc., Redmon, WA, USA) were analyzed using the statistical package SPSS v. 24 (SPSS Inc., Chicago, IL, USA). The data were initially subjected to a normality test using the Kolmogorov–Smirnov test to determine the type of analysis to be performed. The results for individual weights and gene expression were analyzed using a parametric Student’s t-test with a prior Levene’s test to determine the equality of variances. Gene expression data were normalized by performing Log2 of the fold change data. Frequencies were analyzed using contingency tables and the Chi-square statistic, with subsequent adjusted residuals analysis (AR). A significant difference between observed and expected frequencies was considered to be present when *p* < 0.05 and AR < −1.96 or >1.96 to determine more or less observed frequency than expected. Correlations were performed by using Spearman’s test for performances and Pearson’s correlation for molecular biology.

To make the analysis of all the data obtained from each group together more comprehensible, a Discriminant Function analysis was used as a data reduction strategy, taking this technique as appropriate only if Wilks’ lambda had a *p* < 0.05 and the first two functions (in case there was more than one) explained more than 75% of the variance. The Wilks’ lambda test allows testing the null hypothesis that the multivariate means of the groups (centroids) are equal. A group membership assignment analysis and a dot plot were performed to graphically represent the separation of groups when at least two canonical functions were obtained.

In all cases a significant difference was considered when *p* < 0.05.

## 3. Results

### 3.1. Performances

The data obtained for the growth parameters are shown in Figure 2.

A significantly higher weight was observed at all weighting moments for CON groups. Interestingly, the control group grew significantly faster during the growing phase (*p* < 0.0001), while the HC group grew significantly faster during the finishing phase (*p* = 0.047). This compensatory growth resulted in a lack of differences in growth when analyzing the whole period. The same phenomenon can be observed in the weight gain, so that in the end there was no difference in either weight gain or growth. The W1 for HC group at first weighting was 79.4% of CON group, at second weighting was 86.3% and at 3rd weighting was 92.9% of CON weight, respectively.

The coefficient of variation for WG was decreasing over the weightings and was lower for HC group in the second and 3rd weights (20.1% vs. 17.25% for CON and HC in first weight; 19.86% vs. 5.67% in second weight and 19.06% vs. 13.23% in third weighting).

The FCR and ADFI significantly differed in growing and finishing, respectively. The FCRT and ADFIT did not show significant differences between groups.

Considering the differences in average daily intake, FCR and the average price of growing and fattening feed, a difference in production cost of EUR 4.67 per pig has been observed, which would mean a reduction in cost per kilo of fattening of EUR 0.05. However, since the animals entered with different initial weights and this can be a disturbing factor in the calculation of feed conversion rate and average daily gain, the FCR and ADG were standardized to a period of 20–110 kg, using the correction formula that normally use the productive company to compare results, as follows:FCR_norm_ = FCR + ((110 – Final weight) × weight correction_1_) – ((20 – Initial weight) × weight correction_2_))(1)

The weight correction_1_ applied was 9 g for finishers with final weight for 100–110 and 11 g. for those reaching up to 110–120 kg of slaughter weight. The weight correction_2_ was 7 g per kg. Then, the FCRT_norm_ was 2.632 for CON group and 2.623 for HC group, being the difference not significant. The cost of feed using the FCR_norm_ would be EUR 63.01 for CON and 60.91 for HC group. So, in this case the difference in feed cost would be EUR 2.1 less for HC group.

### 3.2. Clinical Observation

The mortality was 1.57% (*n* = 8) for CON and 1.96% (*n* = 10) for HC, where the difference is not significant. The piggery 2 was collectively medicated with doxycycline and bromhexine on day 60 (by water, VO, 5 days) since respiratory symptoms (dyspnea, coughing and sneezing was observed in more than 30% of the animals). Regarding individual treatments, the ratio treatments/animals were 0.13 and 0.22 for CON and HC, respectively. 

With regard to fecal score observations, the frequencies are shown in Figure 3, sorted by growth and finishing phases.

The HC groups showed a significantly higher frequency for the scores “normal” (0) and pasty (1) than expected (AR = 2 and AR = 2,5; *p* = 0.009). There was no difference between expected and observed frequencies during finishing. In this period, practically no score 3 was observed.

### 3.3. Biomarkers for Intestinal Integrity and Immune Stimulation

#### 3.3.1. First Fecal Sampling

The biomarkers showing significant differences are shown in Figure 4.

There was an increased quantitation in group HC for OCL, IFN-α, IFN-γ, IL12-p40, IL1-α, IL6, TNF-α and TGF-β. The cytokines increased are involved in innate cells activation (IFN-α, IL6, IL1-α and TNF-α), cell mediated immunity against intracellular pathogens including activation of macrophages, NK cells and maintenance of Th1 differentiation (IFN-γ) and Th1 induction (IL12-p40). The increased quantity of occluding indicates a good intestinal integrity maintenance.

#### 3.3.2. Second Fecal Sampling

The biomarkers showing significant differences or trends (*p* < 0.1) are shown in Figure 5.

Interestingly, there is a higher gene expression for two of the TJ proteins assayed (CLA and OCL) and a higher gene expression for calprotectin and TNF-α in CON group compared to HC group. There also was a trend to the difference for IL1-α and IL1-β, and IL8. These cytokines are involved in the activation of innate cells and inflammation activation (TNF-α, IL1-α and IL1-β), and recruitment and activation of neutrophils (IL8). A correlation between CAL and IL8 in CON group (*r* = 0.749, *p* > 0.0001) and HC group (*r* = 0.636; *p* = 0.001) was found, as expected since CAL is a protein present in neutrophils.

#### 3.3.3. Abattoir Tissue Samples

The results on tissue samples are shown in Figure 6.

In tissues sampled at slaughter, differences were observed for IL1β, Il12p40 and TNF-α and TGFβ in ileum, while differences were observed for IL12p40, TNF-α and TGFβ in jejunum, always with an increased amount of mRNA in the CON group compared to HC, with an exception for TGF-β. The ratio of TGF/IFN was significantly higher for HC group in jejunum and ileum. Additionally, a trend was observed for increased OCL in the HC group (CON = 2.58 ± 0.31, HC = 3.36 ± 0.33; *p* = 0.093).

### 3.4. Gastric Lesions

The frequency of each gastric observed lesion is showed in Figure 7.

No crateriform or healed ulcers were observed, and the frequency for ulcers was lower than previously observed for other fattening pigs.

There were no significant differences between the observed and expected frequencies for any of the observed lesional categories when comparing the two experimental groups.

### 3.5. Correlation among Performance and Biomarkers

The correlations among biomarkers quantifications and physical performances are shown in Table 7 and Table 8.

Correlations were found between FCR_1 and ZON_1 and treatments, negative between FCR_2 with CLAU_1 and positive with IL1-β_1 and with INF-α_2 and looking at FCRT had positive correlations with CAL_1, ZON_1, TNF-α_1 and the number of individual treatments, and negative with INF-γ_2. Interestingly, there is in any case a positive correlation with ZON and FCR gene expression, which would indicate that overexpression of this TJ protein occurs in animals that perform worse. There is also a correlation between one of the main cytokines related to intestinal immune activation, TNF-α, and obviously with the treatments, which indicates that a poorer health status induces a higher FCR, as expected.

ADFI_1 showed negative correlations with IL10_1 and IL1-α_1, and positive between CAL_2, CLAU_2, IFNα, IL1α, IL1β and TNF-α. ADFI2 showed negative correlations with IL1-α_1, individual treatments, and positive correlations with CAL_2, CLAU_2, IL1-α_ and IL1-β_2. Finally, ADFIT showed negative correlations with CLAU_1 and IL10_1 and positive correlations with IL12p35_1, CAL_2, CLAU_2, INF-α_2, IL12p35_2, IL1-α_2, IL1-β_2 and TNF-α2.

On the other hand, W1 had negative correlations with CLAU_1, IL10_1, INF-γ_2 and IL6_2, and positive correlations with IL1-β_1, IL8_1 and INF-α_2.

W2 showed negative correlations with CLAU_1, IL10_1 and IL6_1 and positive correlations with IL1-β_1, INF-α_2, IL1-α_2 and TNF-α_2. Finally, W3 showed negative correlations with IL12p40_1 and positive correlations with IL12p35_2.

Regarding growth, ADG_1 showed a negative correlation with CLAU_1, IL10_1, IL1-α_1, and positive with CLAU_2, CAL_2, IL1-α_2, IL1-β_2, and TNF-α. ADG_2 and ADGT showed significant negative correlation with IL12p40_1. The fact that the only cytokine correlated with ADGT is IL12 may indicate that mRNA quantification for this cytokine in growth is strongly influencing ADGT from the earliest stages of growth-finishing.

### 3.6. Data Reduction

Since the significance of the Wilks’ lambda test was *p* = 0.393 and *p* = 0.887 for the 63rd and 116th days in feed feces sampling, a discriminant analysis was not applicable. However, for the tissue samples taken at the abattoir, *p* < 0.0001 was obtained for Wilks’ lambda and an allocation capacity of 100% in both ileum and colon. This indicates a distinguishable effect in each of the groups.

Interestingly, when discriminant analysis is performed using only the biomarkers of intestinal integrity (CAL, OCL, CLAU and ZON), a significance for Wilks’ lambda *p* = 0.211 and *p* = 0.234 are obtained for jejunum and ileum, respectively. However, when immunity markers are used, the significance is *p* < 0.0001 for both tissues with 100% of sample group membership assignment capacity. These results suggest that the most obvious effect is on cytokines and not on biomarkers related to gut integrity.

## 4. Discussion

This work describes a better productive performance of the animals that have taken HC with the diet, especially considering that the feed had a reduction of 40 Kcal/kg diet NE. The use of β-mannanase, alone or in combination with other enzymes, has proven useful in improving piglet performance [14], the use of lower quality raw materials [32], prevention of post-weaning diarrhea [33] or even the improvement of pollutant emissions [33,34] in pigs, chickens, laying hens and even dairy cows [35]. Interestingly, in the case of pigs, most of the information available in the scientific literature relates to piglets after weaning, but there is a clear lack of information regarding the use of these enzymes throughout the fattening period and their effect on productive performances and local immune stimulation. To the best of our knowledge, this is the first growing-finishing study under commercial conditions to be documented, and this is of great interest as it replicates the real conditions under which the enzyme has to prove its worth.

The feces score showed an interesting result, since there was a significant difference between expected and observed frequency for the scores “normal” and “pasty” with increased frequency in HC group during growing. Obviously, this difference did not exist in finishing since it was not expected to have diarrhea problems so late. The inclusion of β-mannanase reduces the viscosity of intestinal contents and promotes digestive intercourse, resulting in improved intestinal health and stool quality. It has been shown that the presence of NSP can lead to increased viscosity and decreased interaction between substrates and digestive enzymes [36], leading to increased frequency of abnormal stools. The control of diarrhea by the use of β-mannanase has already been described in weaned piglets [37]. However, there are no reports on the usefulness of this enzyme in fattening animals for this task. In this study, there have been no serious clinical problems related to enteric problems and in fact, although liquid feces were observed in some pens, they did not require treatment of the animals and there were only two animals injected for these symptoms in the HC group and none in the control. It must be recognized that the animals involved in the trial did not have a continuous history of diarrhea at the start of fattening. With regard to stomach lesions, the prevalence of gastric ulcer was lower than that observed in another Spanish study [31], but it must be taken into account that the feed was offered as meal, which is a factor that reduces the risk of gastric ulcer. It should be noted that no chronic or healed ulcers were observed in either experimental group.

In this experiment, we observed a bimodal behavior in ADG; while during the growth phase it was significantly higher in the CON group, at the end it was significantly higher in the HC group, resulting in a total growth that did not differ between groups. An improvement in ADG and ADFI had already been described using mannanase in the presence of corn distillers dried grain with solubles (DDGS), with an increase in ADG proportional to the amount of enzyme added, but not in ADFI [38]. 

On reduced EN diets supplemented with β-mannanase, animals have been shown to achieve at least the same growth performance and feed conversion as animals eating a higher energy feed [39]. In our case, there were no significant differences for ADGT and FCRT, but we found differences sorting by phases. Thus, during the growth phase the CON animals showed significantly better growth while in finishing it was the HC animals that had significantly better growth. The lack of difference in growth, comparing diets with β-mannanase and a reduction of 75 Kcal/kg NE, had already been described for piglets in nursery [40]. Another important finding is the decrease in WG CV throughout the growth-finishing in the HC group; while in the CON group it remained at about 20%. This greater homogeneity in weight gain may mean a better use of the resources present in the feed.

The mechanism that causes animals with a lower energy content to produce at least as much as animals with 75–150 Kcal increases over those treated with β-mannanase is not fully elucidated. It has been determined that there is no difference in nutrient digestibility so it can be assumed that the ME concentration does not change due to the action of the enzyme [41]. However, it has been found that the addition of the enzyme can result in an energy increase of up to 100 Kcal/kg [41].

One of the findings of this study is the difference in ADFI during the growth period. In a meta-analysis on the effects of β-mannanase in different species, Kiarie et al. [42] note that there is no difference for this parameter. The same authors observed a mean difference of 10.8 g/d, while in our study a difference of 74 g/d on behalf of CON in the growing period and 43 g/d for HC in the finishing period was observed. Higher energy diets have been shown to increase daily feed intake. Lv et al. [39] observed in a trial comparing diets with increased energy and with β-mannanase included how higher energy groups had higher feed daily intake, as was observed in this study. Another interesting aspect of our research was the correlation found between ADFI and molecular parameters. The negative correlation between ADFI1 and IL10 could indicate that anti-inflammatory activity is related with decreased voluntary feed intake, since there is energy saving. Obviously, the positive correlation with CAL2, IFNα2, IL1α2, IL1β2 and TNF-α2 would indicate the opposite; a pro-inflammatory state would promote an increase in intake due to the necessary use of energy in that state. Taking the whole period; ADFIT showed negative correlations with CLAU1, IL10 and IL1 α and positive correlations with IL12p351, CAL2, CLAU2, INF α 2, IL12p352, IL1 α 2, IL1β2 and TNF-α, which would indicate that an increased CLAU level and an anti-inflammatory state promotes a reduction in voluntary intake. Similarly, positive correlations indicate that a pro-inflammatory state produces the opposite effect, increasing the feed intake. This increase in feed intake related to an increase in the amount of mRNA for proinflammatory cytotoxins and CAL, an indicator of the presence of macrophages and neutrophils in the intestine, could be related to the chronic inflammatory state promoted by mannans. It has been previously described that the use of mannanase decreases ADFI [38].

Regarding FCR, there was a significant difference in finishing in favor of the HC group. Interestingly, we have found a positive correlation between ZON1 and FCR1 and FCRT. Increased quantities of ZON used to be identified as positive; however, in humans, an upregulation of ZON is related to intestinal inflammation and loss of intestinal integrity [43]. However, in most cases, this term has been investigated by looking at blood protein and not gene expression. The explanation of the IFN-α1 and CAL1 correlation with FCRT is related with the leaky intestine syndrome since IFN-α is a main actor in the syndrome [44]. There is now a consensus that an optimal level of growth-finishing production can be influenced by any degree of activation of the immune system associated with the digestive tract [45]. Moreover, it has been described as the main pro-inflammatory cytokines in intestine as TNF-α, IL-6, and IL1-β [46] and Groschwitz et al. [44] found that TNF-α and IFN-γ were increased in those animals with poor intestinal integrity and affected by chronic intestinal inflammation. Huntley et al. [40] found an increased quantity of seric TNF-α, IL-6, and IL-1β in pigs immunostimulated with *E. coli* LPS in several challenges. 

In this work, we found an increased quantification for IL1-β and TNF-α in the second fecal sampling and at slaughter in ileum. Moreover, a significant increase in IFN-α, TNF-α, and IL-8, was observed in CON pigs comparing the first and the second fecal sampling. On the other hand, a significant decrease for IL6 mRNA quantification was observed for HC group. Since the pro-inflammatory cytokines shift metabolism away from anabolic processes toward a more catabolic state to generate aminoacids and energy necessary to support a chronic immune stimulation, this could be one explanation for the worse performances in terms of FCR and ADG in CON group, compared to HC. 

Interestingly, when data reduction is performed, it is observed that the samples from the CON and HC groups are perfectly differentiable with a Discriminant Function, but only when taking the biomarkers related to immune stimulation, mainly by the activation status of IL1-β and TNF-α in the CON group and TGF-β in the HC group. The fact that the biomarkers of intestinal integrity (ZON, OCL and CLAU) do not allow for group discrimination may indicate that while the effect of β-mannanase addition has a more prolonged effect on cytokines, there is some kind of compaction effect on TJ proteins.

One of the limitations of this study is that it has been carried out under commercial conditions, as the aim was to demonstrate the goodness of β-mannanase use under conditions as similar as possible to those that will be encountered on a production farm. This has led to an initial weight difference that has obviously been maintained throughout the fattening period. However, both groups made the same kilos during fattening, which shows that even with a reduction in net energy content, the use of the enzyme provides an energy digestibility that more than compensates for this reduction. In this trial, the saving in the cost of feed, the reduction in the conversion rate and a growth similar to that of the CON group, has produced a saving of EUR 0.005 per kilo of carcass (EUR 4.7 per pig) using the raw data or 0.0023 EUR/kg using the FCR_norm_.; this saving represents a very important productive and economic advantage, and demonstrates the effectiveness of the use of enzymes for the digestion of substrates that would otherwise be anti-nutritional factors. Not only that, but at the present time with greatly increased feed costs, it allows the inclusion of raw materials that, due to their β-mannan content, cannot otherwise be used, or at least their quantity would be limited.

## 5. Conclusions

In conclusion, the addition of 1–4-endo-D-β--mannanase to the feed throughout the fattening period, in a feed with a 1.6% reduction in net energy compared to the control, allowed the animals to perform as well as the animals on the higher energy diet. Additionally, less frequent diarrhea was observed in the growing phase and less local immune stimulation at slaughter for HC group, probably due to an increase in anti-inflammatory cytokines. All this resulted in a better economic performance in the treated animals, in the order of EUR 4.7 per pig (0.005 EUR/kg of carcass) for this case or EUR 2.1 per pig normalizing the initial weight. In any case, it represents a clear economic advantage over the control group. It should be noted that the trial has been done under commercial conditions, so the results need further investigation.

## Figures and Tables

**Figure 1 animals-12-03012-f001:**
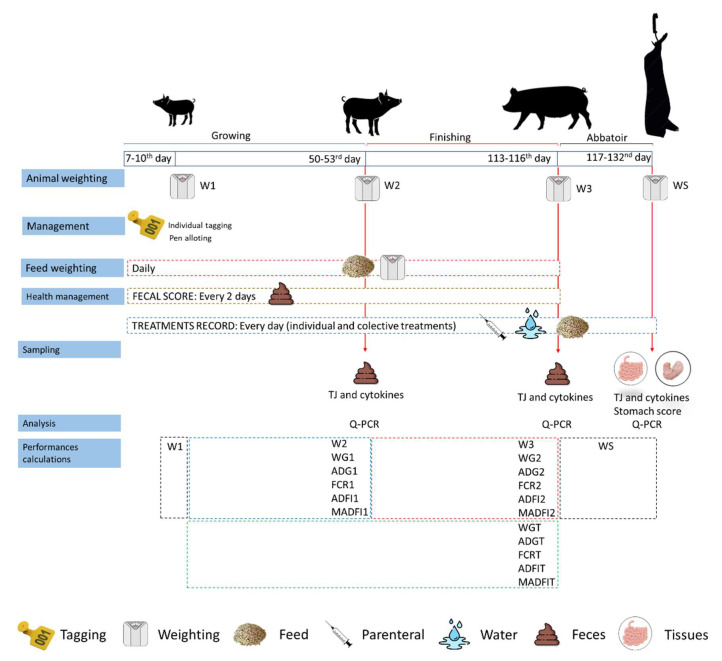
Summary of the methods used in the case–control test, including specific handling, animal and feed weighing, fecal and tissue sampling, monitoring of clinical status, laboratory tests developed, and performance parameters calculated.

**Figure 2 animals-12-03012-f002:**
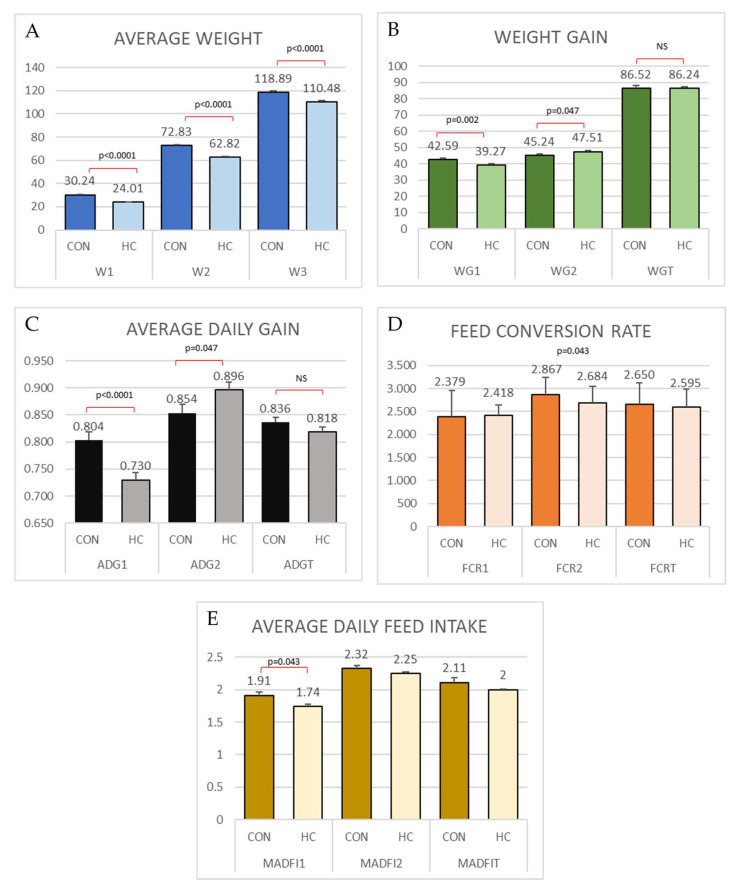
Physical performances. (**A**) Weight (mean ± SEM) at each weighting point, (**B**) Weight gain in growing (WG1), finishing (WG2) and the whole period (WGT). (**C**) Average daily gain in growing (ADG1), finishing (ADG2) and the whole period (ADGT), (**D**) Feed Conversion Rate in growing (FCR1), finishing (FCR2) and the whole period (FCRT), and (**E**) Average Daily Feed Intake in growing (ADFI1), finishing (ADFI2) and the whole period (ADFIT) on individual basis. The bars represent Mean ± SEM. The *p*-value is indicated in those parameters that showed significant differences.

**Figure 3 animals-12-03012-f003:**
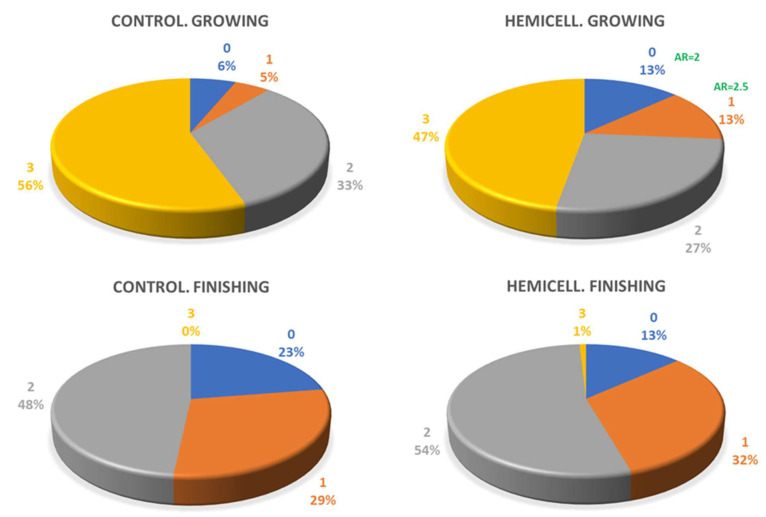
Frequencies observed for fecal consistency score during growing and finishing. The values indicate frequency of each score and AR= adjusted residues in the frequency analysis by contingency tables. Only those frequencies showing AR differed significantly from expected frequency. The frequency difference in the Hemicell-growing group had a significance of *p* = 0.008.

**Figure 4 animals-12-03012-f004:**
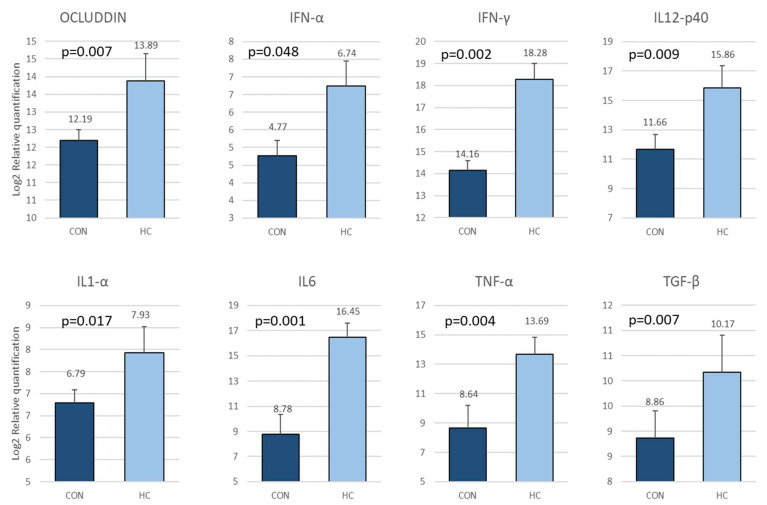
Biomarkers showing differences between CON and HC, expressed as Log2 of relative quantitation on fecal samples at first sampling. The results are expressed as Log2 of fold change, normalizing against the lower value which was assigned as 1. The bars represent Mean ± SEM.

**Figure 5 animals-12-03012-f005:**
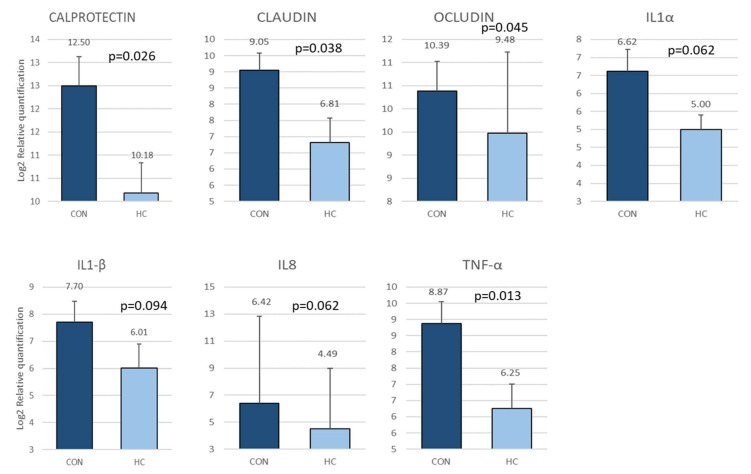
Biomarkers showing differences between CON and HC, expressed as Log2 of relative quantitation on feces in the second sampling. The results are expressed as Log2 of fold change, normalizing against the lower value which was assigned as 1. The bars represent Mean ± SEM.

**Figure 6 animals-12-03012-f006:**
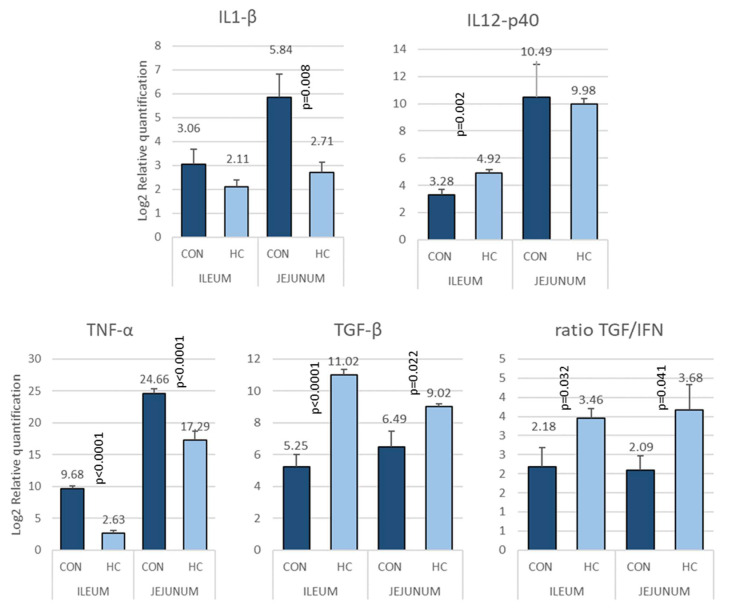
Biomarkers showing differences between CON and HC, expressed as Log2 of relative quantitation on tissue samples at slaughter. The results are expressed as Log2 of fold change, normalizing against the lower value which was assigned as 1. The bars represent Mean ± SEM.

**Figure 7 animals-12-03012-f007:**
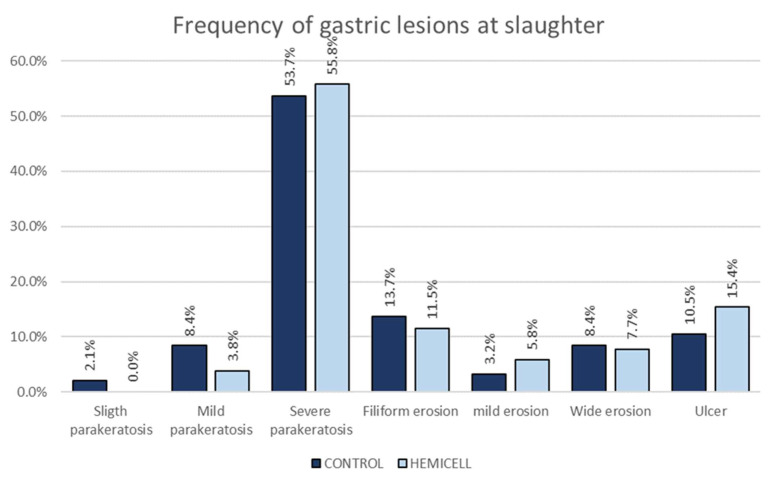
Frequency for each gastric lesional category at slaughter. There were no significant differences in any of the categories between observed and expected frequencies.

**Table 1 animals-12-03012-t001:** Feed composition for growing and finishing.

	Growing		Finishing	
	Control	Treated	Control	Treated
Rawstaff	%	%	%	%
Barley 2C 9,6 PB	23.9	24.86	23.32	24.27
Corn	52	52	55	55
Soybean meal 46	19.4	19.24	19.79	17.58
Pork fat 3–5	1.8	1.03	1.47	0.69
Calcium carbonate	0.69	0.69	0.74	0.74
Monocalcium phosphate hydr.	0.53	0.53	0.33	0.33
Mine salt 96 (NaCl)	0.5	0.5	0.45	0.45
DL- methionine 99	0.14	0.14	0.09	0.09
L-lysine HCL 78	0.43	0.43	0.34	0.35
L-treonine	0.16	0.16	0.11	0.11
L-tryptophan	0.04	0.04	0.04	0.04
L-valine	0.04	0.04		
Hemicell *	0	0.03	0	0.03

* Hemicell HT dry contains endo-1,4-β-mannanase at 160 MU/kg and the dose added assures 48,000 U/kg of final feed.

**Table 2 animals-12-03012-t002:** Analytical values for growing and finishing feeds for each group.

	Growing		Finishing	
	Control	Treated	Control	Treated
Dry matter %	87.94	87.86	87.8	87.72
Raw ashes %	4.15	4.15	3.9	3.91
Raw Protein %	15.9	15.9	15.2	15.2
Raw Fat %	4.27	3.49	3.99	3.21
Raw Fiber%	3.11	3.14	3.02	3.06
NDF %	10.85	11.01	10.82	10.97
ADF %	3.82	3.86	3.73	3.77
Starch %	45.72	46.23	47.33	47.83
Sugars %	2.57	2.57	2.5	2.5
C18:1 %	1.27	0.94	1.15	0.81
C18:2 %	1.34	2.57	1.34	1.26
Calcio %	0.65	0.65	0.63	0.63
Phosforus %	0.58	0.58	0.53	0.53
Pdig porc %	0.3	0.3	0.26	0.26
Sodium %	0.21	0.21	0.19	0.19
Chlorine %	0.44	0.44	0.39	0.39
ED porc kcal/kg (MJ/kg)	3479 (14.56)	3436.81 (14.39)	3464.34 (14.50)	3422.47 (14.33)
EM porc kcal/kg	3351 (14.03)	3308.73(13.85)	3342 (13.99)	3300 (13.99)
EN porc kcal/kg	2540 (10.63)	2500 (10.47)	2540 (10.63	2500 (10.47)
Lysine %	1.1	1.1	1	1
Methionine %	0.39	0.39	0.33	0.33
Methionine + Cistein %	0.67	0.67	0.61	0.61
Threonine %	0.73	0.73	0.67	0.67
Tryptophan %	0.23	0.23	0.21	0.21
Isoleucine %	0.64	0.64	0.61	0.61
Valine %	0.78	0.78	0.72	0.72
Lysine DIS %	1	1	0.9	0.9
Methionine DIS %	0.36	0.36	0.31	0.31
Met. + Cis. DIS %	0.6	0.6	0.54	0.54
Threonine DIS %	0.65	0.65	0.58	0.58
Tryptophan DIS %	0.2	0.2	0.19	0.19
Isoleucine DIS %	0.55	0.55	0.53	0.53
Valine DIS %	0.69	0.68	0.62	0.62
Cost	0.26 EUR/kg	0.25 EUR/kg	0.27 EUR/kg	0.26 EUR/kg

Where NDF = neutral detergent fibre, ADF = Acid-detergent fibre, Pdig porc = percentage of digestible P, ED = digestible energy, EM = metabolizable energy, EN = net energy, DIS = available, MJ = Mega Joules.

**Table 3 animals-12-03012-t003:** Animals weighted at each sampling.

Piggery	Weight 1 (kg)	Days in Feed	Weight 2 (kg)	Days in Feed	Weight 3 (kg)	Days in Feed
	04/10/2021		26/11/2021		18/1/2022	
1 (CON)	113	10	110 *	63	107	116
2 (HC)	112	7	109 *	60	105 **	113

* 3 animals at each group were moved to infirmary pen or died, and the weight was interpolated using the average daily gain of the pen and the days in feed since the last weight. ** 2 animals at each group were moved to infirmary pen or died, and the weight was interpolated using the average daily gain of the pen and the days in feed since the last weight.

**Table 6 animals-12-03012-t006:** Score for aglandular stomach lesions as previously proposed by [31].

Score	Observation
0	Normal mucosa
1	Parakeratosis affecting less than 25% of surface
2	Parakeratosis affecting 25–50% of surface
3	Parakeratosis affecting 100% of surface
4	Presence of small linear erosions
5	Presence of wider erosions
6	Presence of big erosions
7	Ulcer, including acute, chronic or healed ulcers

**Table 7 animals-12-03012-t007:** Correlations among biomarkers in first fecal sampling and physical performances related to feed efficiency.

	FCR1	FCR2	FCRT	ADFI1	ADFI2	ADFIT	MADFI1	MADFI2	MADFIT
CAL_1	0.667	0.119	**0.738 (*)**	−0.452	–0.643	–0.476	–0.143	0.119	–0.119
CLAU_1	0.19	**–0.786 (*)**	–0.071	–0.69	–0.643	**–0.738 (*)**	–0.571	−0.333	−0.548
OCLD_1	0.071	−0.214	0.31	−0.238	−0.19	−0.167	−0.048	0.286	0.119
ZON_1	* **0.952 (**)** *	0.31	* **0.881 (**)** *	−0.381	−0.667	−0.476	0	0.19	−0.071
IFNa_1	0.357	−0.238	0.31	−0.095	−0.333	−0.167	0.214	0.19	0.119
IFNg_1	0.381	−0.167	0.167	−0.19	−0.095	−0.238	0.095	0.381	0.167
IL10_1	0.095	−0.69	−0.095	**−0.714 (*)**	−0.667	**−0.738 (*)**	−0.69	−0.5	−0.667
IL12p35_1	−0.214	0.476	0.095	0.69	0.5	**0.714 (*)**	0.667	0.333	0.571
IL12p40_1	−0.143	0.071	−0.405	0.071	0.214	−0.024	−0.333	−0.524	−0.5
IL1a_1	0.524	−0.167	0.548	**−0.738 (*)**	**−0.833 (*)**	**−0.714 (*)**	−0.429	−0.071	−0.31
IL1b_1	0.31	**0.738 (*)**	0.524	0.452	0.214	0.429	0.619	0.452	0.524
IL6_1	0.31	0.548	0.524	0.595	0.262	0.571	**0.810 (*)**	0.643	**0.714 (*)**
IL8_1	0	0.69	0.167	0.595	0.357	0.548	0.476	0.048	0.262
TGFB1_1	0.429	0.071	0.429	−0.119	−0.357	−0.167	0.024	0.048	−0.048
TNFa_1	0.524	0.167	**0.786 (*)**	−0.143	−0.381	−0.119	0.333	0.548	0.405
Treatments	**0.802 (*)**	0.263	**0.719 (*)**	−0.611	**−0.755 (*)**	−0.659	−0.347	−0.072	−0.335
CAL_2	−0.31	0.452	−0.048	**0.833 (*)**	**0.738 (*)**	**0.810 (*)**	0.595	0.238	0.429
CLAU_2	−0.095	0.667	0.024	* **0.929 (**)** *	**0.738 (*)**	* **0.905 (**)** *	**0.833 (*)**	0.452	0.69
OCLD_2	−0.667	−0.048	−0.452	0.381	0.476	0.476	0.071	−0.095	0.119
ZON_2	−0.619	−0.167	−0.333	0.286	0.333	0.333	0	−0.214	−0.048
IFNa_2	0.167	**0.762 (*)**	0.31	**0.833 (*)**	0.548	**0.786 (*)**	* **0.857 (**)** *	0.524	0.69
IFNg_2	−0.69	−0.595	**−0.905 (**)**	0.095	0.286	0.071	−0.333	−0.595	−0.429
IL10_2	−0.179	−0.357	−0.143	−0.286	−0.214	−0.143	−0.214	0.036	0.036
IL12p35_2	−0.086	0.486	0.086	0.714	0.6	**0.829 (*)**	**0.829 (*)**	* **0.943 (**)** *	* **0.943 (**)** *
IL12p40_2	0.095	0.167	−0.19	−0.071	0.071	−0.119	−0.262	−0.238	−0.286
IL1a_2	−0.619	0.119	−0.31	**0.738 (*)**	**0.833 (*)**	**0.810 (*)**	0.524	0.357	0.548
IL1b_2	−0.667	0.214	−0.5	**0.738 (*)**	* **0.881 (**)** *	**0.833 (*)**	0.476	0.286	0.548
IL6_2	0.167	−0.571	−0.071	−0.5	−0.405	−0.548	−0.476	−0.238	−0.452
IL8_2	−0.571	0.286	−0.31	0.524	0.619	0.619	0.214	0.048	0.262
TGFB1_2	−0.548	−0.214	−0.357	−0.048	0	0.048	−0.31	−0.429	−0.262
TNFa_2	−0.333	0.643	−0.071	**0.833 (*)**	0.69	* **0.881 (**)** *	0.667	0.286	0.595

Marked in bold and with (*) indicates that the correlation is significant at the 0.05 level (bilateral) and marked in bold italics and (**) the significance level is 0.01 (bilateral).

**Table 8 animals-12-03012-t008:** Correlations among biomarkers in second fecal sampling and physical performances related to growth and treatments.

	W1	W2	W3	ADG1	ADG2	ADGT	TREATEMENTS
CAL_1	0.024	−0.262	−0.095	−0.571	0.286	−0.048	**0.826 (*)**
CLAU_1	**−0.810 (*)**	* **−0.857 (**)** *	−0.476	**−0.714 (*)**	0.286	−0.095	0.299
OCLD_1	−0.238	−0.095	−0.048	−0.143	0.238	0	0.335
ZON_1	0.238	−0.19	0.024	−0.619	0.262	0.048	* **0.886 (**)** *
IFN-α_1	0.048	−0.048	0.31	−0.286	0.524	0.5	−0.084
IFN-γ_1	0	0	0.214	−0.19	0.262	0.333	0.12
IL10_1	**−0.786 (*)**	* **−0.881 (**)** *	−0.595	**−0.714 (*)**	0.19	−0.262	0.347
IL12p35_1	0.667	0.762 (*)	0.571	0.619	0	0.333	−0.539
IL12p40_1	−0.333	−0.381	**−0.714 (*)**	0.048	* **−0.857 (**)** *	**−0.810 (*)**	0.036
IL1-α_1	−0.167	−0.452	−0.143	**−0.762 (*)**	0.524	0.048	**0.778 (*)**
IL1-β_1	* **0.905 (**)** *	**0.738 (*)**	0.595	0.31	0.071	0.333	−0.06
IL6_1	0.619	0.643	0.643	0.405	0.119	0.429	0.012
IL8_1	**0.714 (*)**	0.571	0.286	0.429	−0.286	0	−0.299
TGF-β1_1	−0.095	−0.238	−0.095	−0.286	0.071	−0.071	0.527
TNF-α_1	0.452	0.31	0.548	−0.238	0.69	0.571	0.311
Treatments	0.108	−0.359	−0.216	**−0.731 (*)**	0.18	−0.18	
CAL_2	0.286	0.524	0.071	**0.762 (*)**	−0.619	−0.262	−0.407
CLAU_2	0.643	0.786 (*)	0.548	**0.810 (*)**	−0.333	0.214	−0.383
OCLD_2	−0.19	0.143	−0.095	0.524	−0.31	−0.214	−0.371
ZON_2	−0.238	0.024	−0.262	0.357	−0.31	−0.333	−0.479
IFN-α_2	**0.738 (*)**	**0.762 (*)**	0.571	0.643	−0.238	0.238	−0.156
IFN-γ_2	**−0.714 (*)**	−0.452	−0.524	0.167	−0.429	−0.381	−0.623
IL10_2	−0.393	−0.179	0.143	−0.107	0.393	0.107	0.252
IL12p35_2	0.543	* **0.943 (**)** *	* **0.943 (**)** *	0.771	0.086	0.657	−0.314
IL12p40_2	−0.143	−0.238	−0.381	−0.048	−0.5	−0.452	0.311
IL1-α_2	0.071	0.548	0.19	* **0.857 (**)** *	−0.381	−0.048	−0.587
IL1-β_2	0.19	0.619	0.333	* **0.905 (**)** *	−0.333	0.095	−0.611
IL6_2	**−0.786 (*)**	**−0.762 (*)**	−0.571	−0.5	−0.048	−0.333	0.419
IL8_2	0.095	0.381	0	0.667	−0.452	−0.262	−0.275
TGF-β1_2	−0.31	−0.19	−0.286	0.071	−0.071	−0.262	−0.18
TNF-α_2	0.667	**0.810 (*)**	0.524	**0.810 (*)**	−0.238	0.19	−0.467

Marked in bold and (*) indicates that the correlation is significant at the 0.05 level (bilateral) and marked in bold italics and (**) the significance level is 0.01 (bilateral).

## Data Availability

Data are not available.

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
