# Peer review of "Effect of β-Mannanase Addition during Whole Pigs Fattening on Production Yields and Intestinal Health"

_animals, 2022, doi:10.3390/ani12213012_

Round 1
Reviewer 1 Report
My comments were taken into account. I have no more comments or questions.
Author Response
Dear reviewer: thank you very much for you comments. They have improved greately the paper.
Thanks.
Guillermo Ramis

Reviewer 2 Report
A very extensive manuscript, concerning various aspects of the influence of the β-mannanase supplementation in pig nutrition. There are typos in the text in some places that would need to be checked and corrected. Moreover:
Line 42 - typing error
Line 59 - "there is been" - grammatical error
Line 118 - "piggery" alone will do
Table 1 - "pork fat" - is it pork fat in the feed?
Table 4 - the text below the table contains information previously contained in the material and methods, so it is an unnecessary repetition (line 262-264)
Figure 2 - the text highlighted in red should not be under the graph, but in the text of the results discussion
Figure 3 - Below the graphs is an overview of the stool rating scale. It was already included in the text, in the methodology (significance of differences)
References require checking if it complies with the requirements (e.g. line spacing)
Author Response
Dear reviewer: thank you very much for the second review of the paper. We have answered every suggestion in detail:
Line 42 - typing error. Corrected
Line 59 - "there is been" - grammatical error. Corrected
Line 118 - "piggery" alone will do. Corrected
Table 1 - "pork fat" - is it pork fat in the feed?. Yes
Table 4 - the text below the table contains information previously contained in the material and methods, so it is an unnecessary repetition (line 262-264). Removed
Figure 2 - the text highlighted in red should not be under the graph, but in the text of the results discussion. We have included the following sentence:
Interestingly, the control group grew significantly faster during the growing phase (p<0.0001), while the HC group grew significantly faster during the finishing phase (p=0.047). This compensatory growth resulted in a lack of differences in growth when analysing the whole period. The same phenomenon can be observed in the weight gain, so that in the end there was no difference in either weight gain or growth
Figure 3 - Below the graphs is an overview of the stool rating scale. It was already included in the text, in the methodology (significance of differences). The fecal score has been removed and the significance of difference included (p=0.008)
References require checking if it complies with the requirements (e.g. line spacing). Reviewed. Interestingly, line spacing was introduced by the Mendely manager, but effectively, it does not meet Animals' requirements. Not only the line spacing but also the program has not respected the formatting of the name of the journals or the year and has changed authors. Now meet these requirements as far as we have reviewed.
Thanks a lot for your revision that has greatly improved our paper.
Guillermo Ramis

Reviewer 3 Report
Please see the attachment
